# SIGHT: Synthesizing Image-Text Conditioned and Geometry-Guided 3D Hand-Object Trajectories

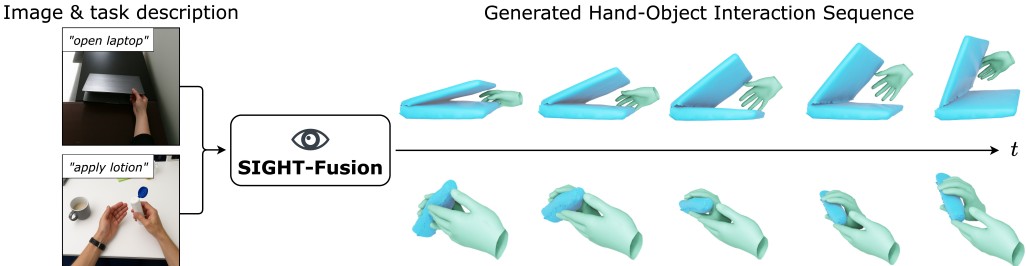

Figure 1: **The proposed SIGHT task and SIGHT-Fusion method.** Given an input image depicting an object being interacted with by one or both hands along with a task description, the newly introduced *SIGHT task* is to generate realistic and physically plausible hand–object motion sequences, mapping out the potential trajectories of the hand(s) during interaction. To address this task, we propose *SIGHT-Fusion*, a diffusion-based method conditioned on both image and text, which integrates a novel *object mesh retrieval* mechanism together with inference-time *physical guidance* to produce realistic and plausible 3D hand–object interaction trajectories.

## Abstract

When humans grasp an object, they naturally form trajectories in their minds to manipulate it for specific tasks. Modeling hand-object interaction priors holds significant potential to advance robotic and embodied AI systems in learning to operate effectively within the physical world. We introduce SIGHT, a novel task focused on generating realistic and physically plausible 3D hand-object interaction trajectories from a single image and a brief language-based task description. Prior work on hand-object trajectory generation typically relies solely on textual input that lacks explicit grounding to the target object, or assumes access to 3D object meshes, which are often considerably more difficult to obtain than 2D images. We propose a novel diffusion-based image-text conditioned generative model that tackles this task by retrieving the most similar 3D object mesh from a database and enforcing geometric hand-object interaction constraints via an inference-time diffusion guidance. We benchmark our model on the HOI4D and H2O datasets, adapting relevant baselines for this novel task. Experiments demonstrate our superior performance in the diversity and quality of generated trajectories, as well as in hand-object interaction geometry metrics. We will publish our code and models.

## 1 Introduction

As our hand grasps an object, we immediately plan out potential maneuvers to manipulate it for our intentions. Consider pouring some juice from a bottle into a cup – it can be as straightforward as rotating your wrist. At a granular level, it requires a continuous adjustment of the hand translation and orientation to transfer exactly the desired amount of liquid into the target receptacle. Humans' hand motion planning systems are remarkably robust in adapting to unseen objects, emulating movements observed from others, and devising paths from visual cues and a downstream task objective in minds.

Robotic agents and AI systems could greatly benefit from a similar capability, whether to anticipate human behavior, generate realistic animations, or facilitate interaction with the physical world.

In this paper, we propose a new task, SIGHT: given a single image showing a hand interacting with an object, along with a brief textual description of the task objective, the goal is to generate realistic and physically plausible 3D hand-object trajectories that meaningfully complete the action initiated in the image. This is a challenging task because the input is highly under-constrained—requiring the model to infer future motion from a single static frame and short text, while ensuring physical plausibility, contact consistency, and semantic alignment with the intended action.

Previous studies on hand-object interactions concentrate on detecting and segmenting pairs of hands and objects interacting within images (Darkhalil et al., 2022; Zhang et al., 2022a), or aim to reconstruct the 3D models of objects and the hands interacting with them from images (Hasson et al., 2019; Ye et al., 2022) or videos (Fan et al., 2024; Ye et al., 2023), but do not generate trajectories. Furthermore, the field of human motion generation has hitherto focused on whole-body motion synthesis (Guo et al., 2022a; 2020b; Tevet et al., 2022b), with little attention paid to synthesizing realistic, task-appropriate and interactive hand motions. In contrast, our focus is on generating dynamic 3D hand-object trajectories, which are crucial for manipulation in everyday human environments, yet using only a single static image and a short task description as inexpensive and easily accessible input.

We propose SIGHT-Fusion, a novel diffusion-based image-text conditioned generative model for tackling this challenging problem. Our framework first extracts textual features representing the task objective and visual features providing the grounded hand-object interaction scenario to condition the generative process. These features are fed into a motion diffusion network trained to produce continuous sequences of 3D hand and object poses, forming complete hand-object interaction trajectories. To encourage smoothness, we augment the standard diffusion objective with a velocity loss during training. At inference time, we introduce a novel geometry-based diffusion guidance mechanism to enforce physically plausible hand-object contact. To alleviate dependence on 3D meshes, we retrieve the most visually similar 3D object mesh to the input image from a mesh database and incorporate a novel interpenetration loss into the test-time denoising process to guide geometry-aware trajectory generation.

We establish comprehensive baselines and evaluation metrics for the newly proposed SIGHT task, leveraging the HOI4D (Liu et al., 2022) and H2O (Kwon et al., 2021) datasets. Our evaluation spans a variety of settings, including different object categories, task types, single- or dual-hand interactions, and both rigid and articulated objects. We adapt several state-of-the-art motion generation methods to our multimodal setting as competitive baselines, and design a suite of quantitative metrics to assess trajectory accuracy, diversity, fidelity, as well as 3D hand-object interaction metrics such as interpenetration and contact consistency. Extensive experiments demonstrate that our approach generates more natural and diverse 3D interaction trajectories with realistic hand-object contacts, outperforming existing methods across both datasets. We further validate key design choices of our system, showing the effectiveness of different feature types, the proposed 3D retrieval augmentation, and the geometry-based diffusion guidance in enhancing trajectory quality.

In summary, the contributions of this paper are: 1) introducing a novel task of generating 3D hand-object interaction trajectories from a single image and a brief textural task description; 2) proposing a new diffusion-based conditional motion generative model that generates diverse and realistic 3D hand-object trajectories given the input text and image conditions; 3) introducing a novel geometry-based diffusion guidance at the inference time given a retrieved 3D object mesh to enforce physically plausible hand-object contact; 4) conducting experiments on two established datasets and demonstrating the superior performance of our proposed method compared to competitive baselines adapted to the new task. We will release our code and model parameters to the public.

## 2 RELATED WORK

**Human motion generation.** Early work on human motion generation has largely focused on motion prediction (Aksan et al., 2021; Cao et al., 2020; Shu et al., 2022; Aliakbarian et al., 2020) and unconstrained motion synthesis (Ling et al., 2020; Wang et al., 2020a;b; Yu et al., 2020; Cai et al., 2021). More recently, there has been growing interest in enhancing controllability in motion generation, leveraging various conditional signals such as text (Karunratanakul et al., 2023; Kim et al., 2023; Tevet et al., 2023a; Zhang et al., 2022b; Dabral et al., 2023), audio (Yi et al., 2023;

Dabral et al., 2023), scene context (Yi et al., 2024), categorical actions (Zhao et al., 2023), goal location (Diomataris et al., 2024) and motion of other people (Liang et al., 2024; Shafir et al., 2024).

Variational Autoencoders (VAEs) have been extensively used for generating realistic human motion from textual descriptions (Petrovich et al., 2022; Zhang et al., 2023a; Yi et al., 2023; Diomataris et al., 2024). Another line of work, such as MotionCLIP (Tevet et al., 2022a) and TM2T (Guo et al., 2022b), employ transformer-based architectures (Vaswani et al., 2017) to align the 3D human motion manifold with the semantically rich CLIP space (Radford et al., 2021), thus inheriting its capabilities.

Building on the success of diffusion models (Ho et al., 2020; Rombach et al., 2021; Dhariwal and Nichol, 2021b) in generative modelling, several diffusion-based methods (Zhang et al., 2022b; Chen et al., 2023; Tevet et al., 2023b; Karunratanakul et al., 2023; Jiang et al., 2024) have been proposed to tackle the task of text-based human motion generation. ReMoDiffuse (Zhang et al., 2023b) proposes a database retrieval mechanism to refine the denoising process. Specifically, the authors retrieve appropriate references from a database in terms of semantic and kinematic similarity and selectively leverage this information to guide the denoising process towards a more high-fidelity result. This inspired the integration of a similarity-based retrieval module in our method. However, the aforementioned methods focus on synthesizing full-body motion, whereas hand-object interaction demands more fine-grained modeling of joint movements and precise alignment with object geometry.

**Hand-object interaction synthesis.** Similarly to 3D human motion synthesis, controllability has become a central objective in hand-object interaction synthesis, aiming to generate physically plausible and natural hand-object motion trajectories conditioned on various modalities. Notably, IMoS (Ghosh et al., 2023) synthesizes full-body motion and 3D object trajectories from text-based instruction labels using conditional variational auto-regressors to model the body part motions in an autoregressive manner. ManipNet (Zhang et al., 2021) proposes a spatial hand-object representation that enables an autoregressive model to predict the hand poses given wrist and object trajectories as input. Given an object geometry, an initial human hand pose as well as a sparse control sequence of object poses, CAMS (Zheng et al., 2023) generates physically realistic hand-object manipulation sequences, using a c-VAE-based motion planner combined with an optimization module. Additionally, ManiDext (Zhang et al., 2024c) adopts a hierarchical diffusion-based approach to synthesize hand-object manipulation based on 3D object trajectories. More recently, BimArt (Zhang et al., 2025) leverages distance-based contact maps to generate realistic bimanual motion, given an articulated 3D object along with its 6 DoF global states and 1 DoF articulation. However, a key limitation of these methods is their reliance on 3D object motion sequences as input, which are often unavailable in real-world generative settings. Another line of work (Christen et al., 2022; Zhang et al., 2024a;b; Jiang et al., 2021) explores the generation of hand-object trajectories, yet requires running simulated environments to train policies using reinforcement learning, and primarily focuses on object grasping rather than modelling the full hand-object interaction.

In contrast to previous works, Text2HOI (Cha et al., 2024) generates 3D motion for hand-object interaction, given a text and canonical object mesh as input. MACS (Shimada et al., 2024) employs cascaded diffusion models to generate object and hand motion trajectories that plausibly adjust based on the object's mass and interaction type. DiffH$_2$O (Christen et al., 2024) further extends this direction by proposing a two-stage temporal diffusion process that decomposes hand-object interaction into distinct grasping and manipulation stages. However, a key limitation of these methods is their reliance on 3D object shapes and their motion sequences as input, which are often unavailable in real-world generative settings. In contrast, our method can generate realistic motion sequences from a single 2D image, without the need for 3D object data.

## 3 METHOD

We start by formulating the SIGHT task, and then describe the framework and loss formulation of the proposed SIGHT-Fusion method (see Figure 2 for an overview), designed to address SIGHT by generating realistic, diverse, and physically plausible hand-object interaction trajectories from a single input image and a brief language-based task description.

### 3.1 TASK DEFINITION

We define the SIGHT task as follows: The input to the hand-object interaction generator $\mathcal{M}$ is an image $\mathcal{I}$ showing a single hand or two hands enacting a certain action on an interacted object, such

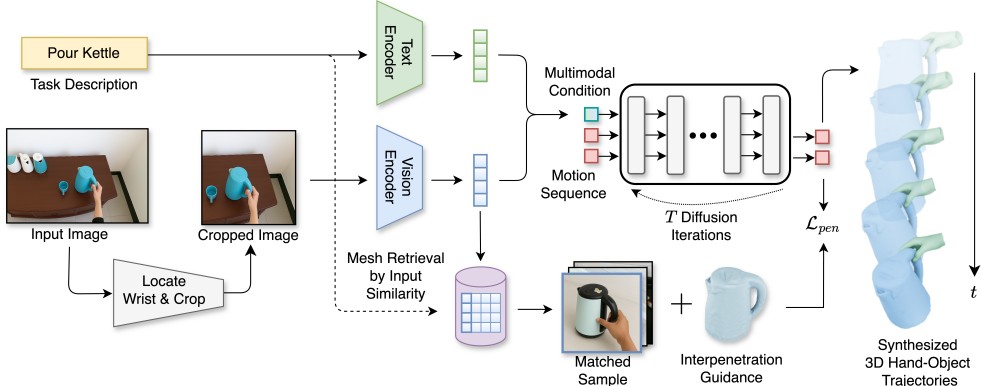

Figure 2: **An overview of SIGHT-Fusion.** Given an input image, we first crop it to a region centered around the hand and the interacted object by detecting the wrist position. We then extract both textual and visual features from the input. These features are passed to a diffusion-based motion generator, which synthesizes realistic and task-appropriate 3D hand-object interaction trajectories. Additionally, we use the visual feature along with the task description to retrieve a corresponding 3D object, which helps provide interpenetration guidance during inference.

as *lifting* or *pouring* a kettle, and a brief textual description $\mathcal{T}$ of that action, consisting of a verb and a noun, e.g. *pour kettle*. The goal is to generate a 3D motion sequence illustrating how the hand(s) and object move from the initial state depicted in the image. Specifically, the expected output is a sequence of hand poses and translations for one or both hands, denoted as $\mathcal{H}^s_{1:F} = \left( \left( \mathcal{H}^s_{P,1}, \mathcal{H}^s_{t,1} \right), \ldots, \left( \mathcal{H}^s_{P,F}, \mathcal{H}^s_{t,F} \right) \right)$ as well as a sequence of object rotations and translations, $\mathcal{O}_{1:F} = \left( \left( \mathcal{O}_{R,1}, \mathcal{O}_{t,1} \right), \ldots, \left( \mathcal{O}_{R,F}, \mathcal{O}_{t,F} \right) \right)$. $F$ is a predefined number of frames and $s \in \{left, right\}$ represents the hand side.

We parameterize $\mathcal{H}^s_{P,i}$ using MANO (Romero et al., 2017) and reason about either a single hand in isolation, or both hands simultaneously for each frame, depending on the scenario. Each hand is parameterized by 16 joint rotations in $\mathcal{H}^s_{P,i} \in \mathbb{R}^{16 \times D}$ and a 3D translation vector in $\mathcal{H}^s_{t,i} \in \mathbb{R}^3$ representing the respective wrist joint's offset from the coordinate origin. We normalize the translation such that the right hand's wrist is always at the origin in the first frame, i.e. $\mathcal{H}^{right}_{t,0} = \overrightarrow{0}_3$. $D$ is dependent on the rotation representation chosen. In our work, we encode all rotations using 6-dimensional representations of rotations, as suggested by (Zhou et al., 2019), yielding $D = 6$. Furthermore, we support reasoning about either rigid objects ($\mathcal{O}_{R,i} \in \mathbb{R}^{F \times D}$, $\mathcal{O}_{t,i} \in \mathbb{R}^{F \times 3}$) or objects with an additional articulated part ($\mathcal{O}_{R,i} \in \mathbb{R}^{F \times 2D}$, $\mathcal{O}_{t,i} \in \mathbb{R}^{F \times 6}$). This high-DoF articulation representation is preferred over explicit articulation parameters to accommodate rigid objects and enable a unified representation for both rigid and articulated objects.

## 3.2 MOTION NETWORK (SIGHT-FUSION)

**Conditioning the trajectory generation.** We aim to provide the motion synthesis model with informative conditioning that helps to disambiguate the intended motion to be generated. While a textual description $\mathcal{T}$ of the task (e.g., *pour kettle*) provides a coarse prior, we argue that the geometry of target object, as conveyed by its visual appearance, is still an important factor for synthesizing realistic trajectories with plausible contacts. Furthermore, we hypothesize that the pose and relative position of the hand in the input image $\mathcal{I}$ offer valuable clues about the starting configuration and initial steps of the motion. Hence, we choose to condition the model on both visual features and the textual description of the task.

Instead of simply providing the model with features from the whole image, we choose to extract features of the object being manipulated while minimizing irrelevant background information. We aim to extract these object features in a manner that is agnostic to the object category. Recognizing that the human hand is the common element across input images in our task, we locate the wrist keypoint using an off-the-shelf model (Pavlakos et al., 2024). We then crop the image to a square region $\mathcal{I}_{crop}$ around the wrist to capture the hand and the object, without explicitly reasoning about

the object itself. See Figure 2 for an example of the result. An analysis of the efficacy and failure cases of the approach is provided in the Supp. Mat.

**Diffusion-based generation.** Given the shared requirement of reasoning about temporal sequences of both hand and object in 3D, we adopt state-of-the-art diffusion models for the SIGHT task of generation of 3D hand-object interaction trajectories. We model the motion synthesis process over a sequence of $F$ frames, each having $J$ joints with $D$ pose features. For example, $J = 2 \times 16 + 1$ represents two hands (16 joints each) and a rigid object, which we simply model as an additional joint. Respecting bone length constraints, we further set $M$ to be the number of joints with translational freedom, e.g. $M = 4$ for a bimanual interaction with an articulated object, where the left and right wrist as well as both object parts have translation components. The synthesis process starts with a randomly sampled noise input $x_T \sim \left( \mathcal{N}(\vec{0}, I_{F \times J \times D}), \mathcal{N}(\vec{0}, I_{F \times M \times 3}) \right)$. Here, $T$ represents the total number of denoising steps to perform. Then a denoiser $\mathcal{G}$ is applied to iteratively denoise $x_T$ into the final motion $x_0$ in the $T$ diffusion steps.

For each input sample $(\mathcal{I}_{crop}, \mathcal{T})$, we extract the text query feature $f_\mathcal{T} = E_\mathcal{T}(\mathcal{T})$ and the visual feature $f_{\mathcal{I}_{crop}} = E_{\mathcal{I}_{crop}}(\mathcal{I}_{crop})$. Here, $E_\mathcal{T}$ denotes the text encoder from the CLIP model (Radford et al., 2021) and $E_{\mathcal{I}_{crop}}$ is a visual encoder based on SigLip (Zhai et al., 2023). Let $c = (f_\mathcal{T}, f_{\mathcal{I}_{crop}})$ be the conditioning information and $\hat{x}_t$ represent the motion sequence generated in the diffusion step $t \in \{1, ..., T\}$, with $x_0$ denoting the original sequence. We initialize $\hat{x}_T$ by sampling random Gaussian noise. During training, we uniformly sample $t \in \{1, ...T\}$ to train $\mathcal{G}$, following DDPM (Ho et al., 2020). During inference, for every step $t$, the denoising model $\mathcal{G}$ denoises $\hat{x}_{t+1}$ into $\hat{x}_t$ given conditioning $c$ and timestep $t$ as additional information. We adopt the Transformer decoder architecture proposed in (Zhang et al., 2023b) for $\mathcal{G}$.

To train our motion generator, we rely on a mean squared error (MSE) loss between the model's output and the ground-truth motion sequence. Specifically, we encode the object's rotation and translation $(\mathcal{O}_{R,i}, \mathcal{O}_{t,i})$ in each frame $i$ relative to the right wrist $(\mathcal{H}_{R,i,0}^{right}, \mathcal{O}_{t,i,0})$, i.e., the 0-th joint as defined by OpenPose (Cao et al., 2017), within the same frame. We also employ a velocity loss to further reduce jitter in the generated sequences. Our full loss formulation $\mathcal{L}$ consists of the DDPM-based simple reconstruction loss $\mathcal{L}_{rec}$ and a velocity loss $\mathcal{L}_{vel}$, which penalizes discrepancies in temporal differences for each joint:

$$\mathcal{L} = \mathcal{L}_{rec} + \lambda_{vel}\mathcal{L}_{vel},$$

$$\mathcal{L}_{rec} = \mathbb{E}_{t \sim \text{Uniform}(\{1,...T\})} \|\hat{x}_t - x_0\|_2^2,$$

$$\mathcal{L}_{vel} = \mathbb{E}_{t \sim \text{Uniform}(\{1,...T\})} \sum_{f=1}^{F-1} \|(\hat{x}_{t,f} - \hat{x}_{t,f-1}) - (x_{0,f} - x_{0,f-1})\|_2^2$$

where $\hat{x}_t = \mathcal{G}(\tilde{x}_t, t, c)$ and $\tilde{x}_t$ is an artificially noised version of the ground-truth motion $x_t$ used to train $\mathcal{G}$. To avoid scenario-specific hyperparameter tuning, we set $\lambda_{vel} = 1$.

**Guiding the generation towards more realistic trajectories.** The method discussed so far relies only on visual information and the prior from training to synthesize realistic trajectories. However, since it only works with a single monocular image, it has limited access to the object's geometry. Moreover, the absence of explicit constraints or loss terms that encourage realism poses additional challenges in generating physically plausible motions.

To aid the model in synthesizing more realistic trajectories, we use diffusion guidance (Dhariwal and Nichol, 2021a) to steer the generation process towards more physically plausible outcomes. Diffusion Guidance provides an effective means of controlling the sample generated by a diffusion model by minimizing a loss acting on the intermediate output of each diffusion step. This approach has been commonly used in text-conditioned image synthesis (Dhariwal and Nichol, 2021a; Epstein et al., 2023). Specifically, at each diffusion step, we minimize a loss $\mathcal{L}_{pen}$ penalizing hand-object interpenetration:

$$\mathcal{L}_{pen} = \sum_{i=1}^{F} \sum_{v \in \mathcal{V}_i \cap \text{Int}(\mathcal{M}_i)} \min_{p \in \mathcal{V}_i(\mathcal{M}_i)} \|v - p\|_2^2,$$

where $\mathcal{M}_i$ is the mesh $\mathcal{M}$ of the interacted object at frame $i \in \{1, ..., F\}$ transformed according to the synthesized object motion $(\mathcal{O}_{R,i}, \mathcal{O}_{t,i})$. $\text{Int}(\mathcal{M}_i)$ denotes the space inside the mesh volume,

and $\mathcal{V}_i(\mathcal{M}_i)$ are the vertices of the mesh. The set of MANO (Romero et al., 2017) hand vertices $\mathcal{V}_i = \mathrm{MANO}(\mathcal{H}_{P,i}, \mathcal{H}_{t,i})$ is from the synthesized hand poses $\mathcal{H}$ in the frame $i$.

Calculating $\mathcal{L}_{pen}$ requires access to the 3D object $\mathcal{M}$ during inference, which is not available as part of our input. We hence explore the strategy of using a database $\mathcal{D}$, where each entry consists of a key $k = (k_{\mathcal{T}}, k_{\mathcal{I}_{crop}})$ formed by a task descriptions $k_{\mathcal{T}}$ and visual features $k_{\mathcal{I}_{crop}}$ from the cropped region of interaction scenes, and a corresponding value $\mathcal{D}_k$, which contains the associated object mesh. The features $k_{\mathcal{I}_{crop}}$ are extracted from an image patch determined by the wrist position, following the same procedure used for the input image.

Given a new interaction defined by visual features $f_{\mathcal{I}_{crop}}$ and a task description $f_{\mathcal{T}}$, we query $\mathcal{D}$ to retrieve the object mesh whose visual features $k_{\mathcal{I}_{crop}}$ are most similar to $f_{\mathcal{T}}$, based on cosine similarity. Formally,

$$\mathcal{M} = \mathcal{D}_{k^*}, \; k^* = \arg\max_{(k_{\mathcal{I}_{crop}}) \in \mathrm{keys}(\mathcal{D})} \frac{k_{\mathcal{I}_{crop}} \cdot f_{\mathcal{I}_{crop}}}{\|k_{\mathcal{I}_{crop}}\| \cdot \|f_{\mathcal{I}_{crop}}\|}$$

Additionally, we further filter the results by measuring the similarity between the retrieved task description $k_{\mathcal{T}}$ and the query task description $f_{\mathcal{T}}$. Note that this inference-time optimization approach is equally sensitive to meshes in the database as well as unseen ones, whereas a learning-based approach might generalize poorly to out-of-distribution objects. We provide experiments regarding the accuracy of our retrieval scheme and the alignment of the retrieved meshes with their visual observations shown depicted in the images in the Supp. Mat.

## 4 EXPERIMENTS

In this section, we evaluate SIGHT-Fusion's generated interaction trajectories with respect to their diversity, realism, and physical plausibility against baseline models on the SIGHT task. Additionally, we demonstrate that visual features provide explicit grounding for interaction scenarios, and both retrieval and inference-time guidance lead to better motion generation.

### 4.1 EXPERIMENTAL SETUP

To establish a comprehensive evaluation for our newly introduced SIGHT task, we adapt two hand-object interaction video datasets for our benchmark. We incorporate established metrics from the whole-body motion generation literature to evaluate the generated hand-object trajectories, and further assess the physical plausibility of the synthesized interactions using metrics measuring hand-object contact and interpenetration.

**Datasets.** The *HOI4D* dataset (Liu et al., 2022) consists of first-person videos of hands interacting with everyday objects from 16 categories. 3D scans of object instances are provided in the dataset, and 3D hand and 6D object poses corresponding to each video sequence are also available. The number of object instances varies within each category, ranging from 31 to 47, and the action tasks associated with these objects vary between 2 to 6 per category. Altogether, the dataset defines 31 action tasks. We merge and rename several tasks to extract 10 *actions* from the original 31 tasks. Details of this grouping are provided in the Supp. Mat. As there is no publicly available dataset split introduced for HOI4D, we define our own *instance split* with the aim of testing cross-object-instance generalization ability for methods addressing the SIGHT task. We split each category action group into one subgroup with instances to use only for the training set, and another subgroup with instances to use only for the test set. Our proposed *instance split* thus allows a meaningful evaluation of cross-instance action knowledge transfer within the same object category. It is also highly suitable for highlighting the benefits of our category-agnostic and mesh-independent method. Simultaneously, more advanced reasoning is required for successful cross-instance transfer, as object instances in the test set may look different from those seen during training. A detailed description of the split is provided in the Supp. Mat.

The *H2O* dataset (Kwon et al., 2021) contains videos that capture two-hand interactions with eight different objects. It includes 36 distinct action classes along with rich annotations for 3D poses of the left and right hands and 6D object poses. 3D object meshes are also provided in the datset. We follow the official action split provided with the dataset.

**Evaluation metrics.** For rigorous benchmarking, our SIGHT task includes both hand trajectory-specific metrics, as well as metrics reasoning about the interaction of the hand and the object. We measure the quality of the hand-object interaction trajectories generated by a method $M$ based on their *accuracy* (ACC), *diversity* (DIV), and *fidelity to ground-truth* (FID). This combination of metrics is commonly used in the human motion generation literature (Guo et al., 2020b; Tevet et al., 2022b), as it measures the quality of motion sequences with respect to several desirable criteria.

To measure the **accuracy** (ACC) of a motion generation method $M$, we consider the fraction of generated interaction trajectories a reference action classifier can correctly assign to the respective actions they were supposed to depict. The **diversity** (DIV) of $M$'s outputs is calculated through the Fréchet Inception distances (FIDs) (Heusel et al., 2017) between action classifier features extracted from generated (gen.) and ground-truth (GT) motion groups. The diversity of the generated trajectories corresponds to the FID of one-half of a trajectory group to the other half. Diversity scores closer to that of the GT trajectories are better. The **fidelity** (FID) of generated hand trajectories is calculated through the Fréchet Inception distances, similar to the diversity metric. Lower FIDs between generated and ground-truth sequences translate to a motion generation method producing trajectories more similar in distribution to real reference trajectories.

We further investigate the quality of the generated hand-object interaction using *interpenetration depth* (ID; in $cm$), interpenetration volume (IV; in $cm^3$) and *contact ratio* (CR). We report the largest **interpenetration depth** (ID), in centimeters, to which the hand(s) penetrated any of the objects. The maximum is taken over all frames of all sequences generated for the validation set. This metric is to be minimized to generate realistic motion sequences. The **interpenetration volume** (IV) measures the volume of the section of the hand penetrating the object, in cubic centimeters and evaluated at the frame of the maximum interpenetration depth. This metric is a volumetric equivalent of the interpenetration depth. We additionally consider the **contact ratio** (CR) of hand vertices closer than some threshold $\tau$ to any vertex of the object, averaged over all frames and sequences. Here, we set $\tau = 5mm$. A contact ratio close to that of the ground truth is desirable, as the grasp is considered to mimic that in the ground-truth sequence more closely.

**Implementation details.** For each run, consistent with the human motion generation literature (Guo et al., 2022a; 2020a; Tevet et al., 2022b), we select the checkpoint achieving the lowest FID metric on the test set, so as to produce the trajectories most similar to the test set trajectories. We average each score reported for HOI4D over 20 evaluations, and each score reported for the bimanual (and computationally more expensive) H2O over 5 evaluations for statistical robustness. Hyperparameters are provided in Supp. Mat.

**Baselines.** We compare our method against three state-of-the-art baselines. As the proposed SIGHT task is novel, we adapt existing baselines from the whole-body motion generation literature.

*MDM* (Tevet et al., 2022b) trains a motion diffusion model using a transformer encoder backbone with classifier-free guidance. As a conditioning signal, it employs a CLIP-based textual embedding, which is processed by the model together with the motion features to be denoised. Following the original formulation, we utilize the simple diffusion objective (Ho et al., 2020) augmented with the geometric losses (i.e., position and velocity) and retain all the standard hyperparameters.

*MotionDiffuse* (Zhang et al., 2022b) is a diffusion-based model for motion generation that softly fuses CLIP encoded text features into the denoising process using cross-attention and stylization blocks. Following the original formulation, we use the simple diffusion objective and retain all the standard hyperparameters.

*ReMoDiffuse* (Zhang et al., 2023b) is a retrieval-augmented motion diffusion model that conditions generation on semantically and kinematically similar motion trajectories. Given a textual description, it retrieves relevant motion samples from a database and feeds them into a semantics-modulated transformer, which extracts informative cues to guide the denoising process. We adopt the original text-conditioned loss formulation and use the standard hyperparameters.

## 4.2 Evaluation of Generated 3D Hand-Object Interaction Trajectories

As evident in Table 1, our method outperforms all baselines in four of six metrics and performs favorably in the remaining two when evaluating on HOI4D. The object-image conditioning provides a strong signal about the object's geometry and its relationship to the hand, resulting in a significantly

Table 1: **Quantitative results on the HOI4D dataset (Liu et al., 2022).** Our method achieves the best performance compared to three state-of-the-art baselines. For comparison purposes, all methods use the same motion length of 196 frames. -I model variants use both the image and text modalities for conditioning. '→' means results are better if the metric is closer to the ground-truth motions. We run all the evaluation 20 times for statistical robustness and '±' indicates the 95% confidence interval. The best results are in **bold** and the second best results in underline.

| Method | ACC ↑ | DIV → | FID ↓ | ID (cm) ↓ | IV (cm³) ↓ | CR → |
|---|---|---|---|---|---|---|
| Ground-Truth | $1.000^{\pm 0.000}$ | $11.782^{\pm 0.162}$ | — | $2.742$ | $14.981$ | $0.063$ |
| MDM | $0.996^{\pm 0.002}$ | $\mathbf{11.773^{\pm 0.211}}$ | $\underline{0.132}^{\pm 0.016}$ | $3.471^{\pm 0.227}$ | $35.145^{\pm 7.117}$ | $0.074^{\pm 0.002}$ |
| MotionDiffuse | $\underline{0.999}^{\pm 0.001}$ | $11.613^{\pm 0.256}$ | $0.137^{\pm 0.018}$ | $3.116^{\pm 0.187}$ | $31.033^{\pm 4.777}$ | $0.074^{\pm 0.003}$ |
| ReMoDiffuse | $0.991^{\pm 0.002}$ | $11.702^{\pm 0.205}$ | $0.238^{\pm 0.031}$ | $4.315^{\pm 0.158}$ | $72.595^{\pm 9.456}$ | $0.109^{\pm 0.003}$ |
| MDM-I | $0.986^{\pm 0.002}$ | $11.653^{\pm 0.188}$ | $0.153^{\pm 0.012}$ | $3.271^{\pm 2.360}$ | $25.286^{\pm 3.828}$ | $0.070^{\pm 0.003}$ |
| MotionDiffuse-I | $1.000^{\pm 0.000}$ | $11.481^{\pm 0.202}$ | $0.144^{\pm 0.015}$ | $2.995^{\pm 0.836}$ | $28.269^{\pm 9.353}$ | $\mathbf{0.068}^{\pm 0.002}$ |
| ReMoDiffuse-I | $1.000^{\pm 0.000}$ | $11.744^{\pm 0.200}$ | $0.099^{\pm 0.008}$ | $3.748^{\pm 0.136}$ | $74.467^{\pm 14.134}$ | $0.079^{\pm 0.003}$ |
| Ours | $\mathbf{1.000}^{\pm 0.000}$ | $\underline{11.820}^{\pm 0.204}$ | $\mathbf{0.078}^{\pm 0.007}$ | $\mathbf{2.928}^{\pm 0.037}$ | $\mathbf{23.108}^{\pm 7.339}$ | $\underline{0.069}^{\pm 0.002}$ |

Table 2: **Quantitative results on the H2O dataset (Kwon et al., 2021).** Our method achieves the best performance on the challenging FID and CR metrics, and comparable performance on the ID and IV metrics, compared with three state-of-the-art baselines. For comparison purposes, all methods use the same motion length of 196 frames. '→' means results are better if the metric is closer to the real motions. We run all the evaluation 5 times for statistical robustness and '±' indicates the 95% confidence interval. The best results are in **bold** and the second best results in underline.

| Method | ACC ↑ | DIV → | FID ↓ | ID (cm) ↓ | IV (cm³) ↓ | CR → |
|---|---|---|---|---|---|---|
| Ground-Truth | $0.856^{\pm 0.017}$ | $12.397^{\pm 0.403}$ | — | $3.452$ | $13.562$ | $0.084$ |
| MDM | $0.866^{\pm 0.021}$ | $\underline{12.595}^{\pm 0.330}$ | $0.104^{\pm 0.005}$ | $3.729^{\pm 2.941}$ | $\mathbf{17.455}^{\pm 7.868}$ | $\underline{0.112}^{\pm 0.007}$ |
| MotionDiffuse | $\mathbf{0.884}^{\pm 0.029}$ | $12.689^{\pm 0.467}$ | $0.111^{\pm 0.022}$ | $\mathbf{3.542}^{\pm 0.231}$ | $36.041^{\pm 21.192}$ | $0.122^{\pm 0.002}$ |
| ReMoDiffuse | $\underline{0.879}^{\pm 0.024}$ | $\mathbf{12.514}^{\pm 0.117}$ | $\underline{0.100}^{\pm 0.002}$ | $4.036^{\pm 0.075}$ | $21.035^{\pm 8.233}$ | $0.118^{\pm 0.010}$ |
| Ours | $0.869^{\pm 0.008}$ | $12.810^{\pm 0.566}$ | $\mathbf{0.087}^{\pm 0.012}$ | $\underline{3.618}^{\pm 0.113}$ | $\underline{20.111}^{\pm 5.172}$ | $\mathbf{0.092}^{\pm 0.005}$ |

lower FID than all baselines. The advantages of our method further become evident on the ID, IV, and CD metrics, where our guidance term helps us achieve natural, low-penetration grasps.

As H2O includes only one object instance per task across its 8 tasks, models can rely on strong priors learned from the ground-truth sequences without needing to reason about object geometry. As shown in Table 2, although our method cannot unfold its full potential in such a setting, it still achieves the best average performance, outperforming all baselines strongly in FID and Contact Ratio while being the runner-up on Interpenetration Depth and Volume. We attribute this to the evaluation on H2O favoring methods that learn to reproduce training trajectories over those that reason about the object.

Examples of single-handed (HOI4D) and bimanual (H2O) trajectories generated by our models, as well as comparisons to the ReMoDiffuse (Zhang et al., 2023b) and MDM (Tevet et al., 2022b) baselines, are visualized in Figure 3, confirming that our method achieves more physically realistic results while maintaining superior quantitative performance. More qualitative results and baseline comparisons are provided in Supp. Mat.

**Ablations.** We present an ablation of several conditioning strategies for our model in Table 3, starting with the text-based baseline (Zhang et al., 2023b). Generating motions based only on text results in a lower accuracy while also markedly worsening the FID. Using whole-scene features by themselves improves these two metrics, with the FID reducing to less than half of the original value. Considering the object in isolation gives good accuracy and diversity, while slightly worsening the FID. Combining text with scene or object features gives us better FID and accuracy values than reasoning about them in isolation. Adding our guidance term to the model that combines text features with object features restores our method, giving the best performance on all metrics.

We further investigate different versions of our guidance term in Table 4, varying between using the ground-truth mesh, random meshes drawn from the training set and in the same category as the ground truth, as well as using meshes retrieved based on our proposed visual matching scheme. The results validate the need for the guidance term and further validate the usefulness of our feature-based matching over simply choosing a random mesh. Notably, while our guidance term modifies the generated motions to reduce physical artifacts, it does not worsen their trajectory-related metrics (ACC, DIV, FID), showing its usefulness for the investigated task.

Table 3: **Ablation study on different conditioning signals.** Adding our proposed guidance term to the model, and combining text with object features yields the best performance on all metrics.

| Method | ACC ↓ | DIV → | FID ↓ | IV (cm$^3$) ↓ | CR → |
|---|---|---|---|---|---|
| Ground-Truth | $1.000^{\pm 0.000}$ | $11.782^{\pm 0.162}$ | — | $14.981$ | $0.063$ |
| Text | $0.991^{\pm 0.002}$ | $11.702^{\pm 0.205}$ | $0.238^{\pm 0.031}$ | $72.595^{\pm 9.456}$ | $0.109^{\pm 0.003}$ |
| Scene Img. | $\underline{0.995}^{\pm 0.002}$ | $11.684^{\pm 0.189}$ | $0.106^{\pm 0.011}$ | $77.603^{\pm 13.352}$ | $0.074^{\pm 0.002}$ |
| Object Img. | $\mathbf{1.000}^{\pm 0.000}$ | $\underline{11.831}^{\pm 0.198}$ | $0.129^{\pm 0.012}$ | $82.878^{\pm 10.054}$ | $0.081^{\pm 0.002}$ |
| Text + Scene Img. | $\mathbf{1.000}^{\pm 0.000}$ | $\mathbf{11.744}^{\pm 0.200}$ | $0.099^{\pm 0.002}$ | $74.467^{\pm 14.134}$ | $0.079^{\pm 0.003}$ |
| Text + Object Img. | $\mathbf{1.000}^{\pm 0.000}$ | $11.892^{\pm 0.197}$ | $\underline{0.079}^{\pm 0.008}$ | $\underline{30.773}^{\pm 9.113}$ | $\underline{0.071}^{\pm 0.005}$ |
| Text + Object Img. + Guidance (Ours) | $\mathbf{1.000}^{\pm 0.000}$ | $11.820^{\pm 0.204}$ | $\mathbf{0.078}^{\pm 0.007}$ | $\mathbf{23.108}^{\pm 7.339}$ | $\mathbf{0.069}^{\pm 0.002}$ |

Table 4: **Ablation of guidance.** We study the effect of using our guidance term with different mesh types and validate its effectiveness in reducing interpenetration while letting the motions maintain a high diversity and fidelty to the ground-truth.

| Method | ACC ↓ | DIV → | FID ↓ | IV (cm$^3$) ↓ | CR → |
|---|---|---|---|---|---|
| Ground-Truth | $1.000^{\pm 0.000}$ | $11.782^{\pm 0.162}$ | — | $14.981$ | $0.063$ |
| True Mesh | $1.000^{\pm 0.000}$ | $11.589^{\pm 0.152}$ | $0.076^{\pm 0.005}$ | $21.207^{\pm 6.234}$ | $0.069^{\pm 0.004}$ |
| No Guidance | $\mathbf{1.000}^{\pm 0.000}$ | $11.892^{\pm 0.197}$ | $\underline{0.079}^{\pm 0.008}$ | $30.773^{\pm 9.113}$ | $0.071^{\pm 0.005}$ |
| Rand. Mesh in Category | $\mathbf{1.000}^{\pm 0.000}$ | $\mathbf{11.802}^{\pm 0.227}$ | $\mathbf{0.078}^{\pm 0.007}$ | $\underline{24.132}^{\pm 7.347}$ | $\underline{0.070}^{\pm 0.002}$ |
| Retrieved Mesh (Ours) | $\mathbf{1.000}^{\pm 0.000}$ | $\underline{11.820}^{\pm 0.204}$ | $\mathbf{0.078}^{\pm 0.007}$ | $\mathbf{23.108}^{\pm 7.339}$ | $\mathbf{0.069}^{\pm 0.002}$ |

Figure 3: **Baseline comparisons.** We provide qualitative comparisons of our method's synthesized motions (top row) to those of ReMoDiffuse (middle row) and MDM (bottom row), on objects from H2O (left, right) and HOI4D (center). The baselines exhibit interpenetration artifacts in the generated sequences, whereas our method produces realistic trajectories.

## 5 DISCUSSION

We introduce a novel task: the generation of natural and diverse 3D hand-object trajectories conditioned on single image inputs and language-based task descriptions. To address this challenging problem, we propose a novel diffusion-based image-text-conditioned generative method SIGHT-Fusion. To further improve the realism of generated trajectories, we integrate a visual similarity-based retrieval mechanism and an interpenetration guidance term during inference. We set up comprehensive baselines adapted to the new task, and report numerous relevant metrics on the HOI4D and H2O datasets. Experimental results demonstrate that our method outperforms baselines in terms of both diversity, naturalism, and physical plausibility. Ablation studies further validate the effectiveness of our proposed retrieval and inference-time guidance strategies and show how different input conditions influence the performance.

**Limitations and future directions.** While SIGHT-Fusion enables realistic and diverse 3D hand-object interaction motion generation, it relies on datasets that provide the corresponding 3D object motions in each frame. This can partially be alleviated through the use of off-the-shelf object pose reconstruction methods (Wen et al., 2024). Integrating existing image-to-3D generation pipelines could further improve the generalization ability of SIGHT-Fusion. Despite incorporating interpenetration guidance, the generated sequences may still exhibit hand-object penetrations as no hard constraints applied to the generated motions. The realism of the generated trajectories can be further improved, e.g. through integration with differentiable physical simulators (Turpin et al., 2023). We hope that our work will invite greater interest in this SIGHT task, and that our generative framework will be of use in related applications, such as action anticipation and object manipulations in robotic settings.

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

# A    APPENDIX

## A.1    MESH RETRIEVAL

Qualitative visualizations of validation set images/meshes, corresponding images/meshes retrieved from the training set for guidance-based optimization, as well as random images/meshes from the same category in the training set that were rejected in favor of the retrieved ones, are shown in Table 5.

Table 5: Qualitative results of mesh retrieval.

| Original image | Original mesh | Retrieved image | Retrieved mesh | Random image | Random mesh |
| --- | --- | --- | --- | --- | --- |

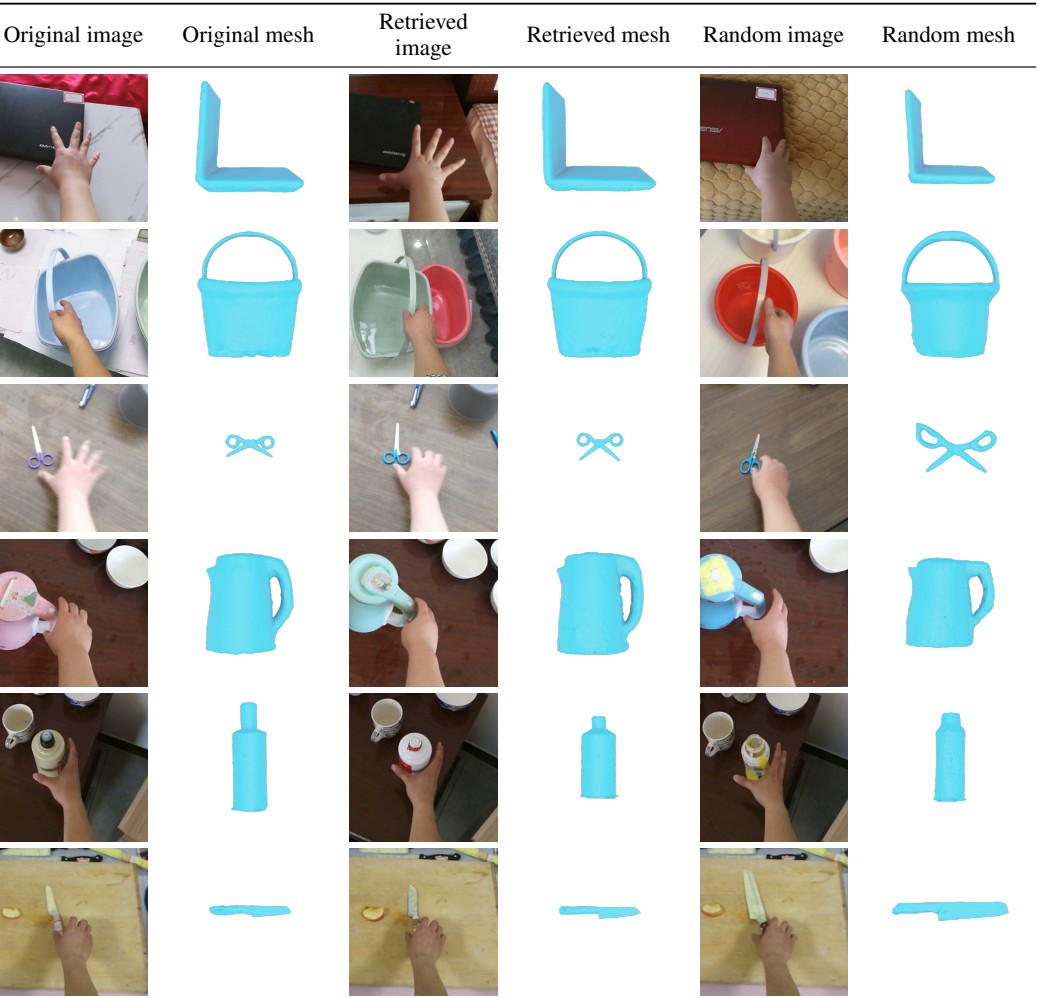

We further provide a comparison of different variants of our proposed mesh retrieval scheme in Table 6. Specifically, we investigate the Chamfer distance to the ground-truth mesh when (a) selecting a random mesh from the same category *(Random)*, (b) selecting the mesh most similar based on visual features from the whole image *(Image)*, as well as (c) selecting the mesh most similar based on visual features from a wrist-centered patch capturing the object *(Object)*. The best performance when using setting (c) confirms the disambiguating property of retrieval based on visual features from a wrist-centered region closely capturing the hand-object interaction.

## A.2    FURTHER QUALITATIVE EXAMPLES

We provide additional visualizations of hand-object interaction trajectories generated by our method in Figure 4. The respective conditioning images are provided in Figure 5. Please also consult the video files attached to this supplementary material for further and richer qualitative results.

Table 6: Chamfer distance to ground-truth mesh, in centimeters, for different query types used to retrieve meshes from the HOI4D training set.

| Setup | CD (cm) ↓ |
|---|---|
| Random | $1.955 \pm 0.066$ |
| Image | $1.718 \pm 0.000$ |
| Object | $\mathbf{1.618} \pm 0.000$ |

## A.3 DATASET SPLIT

We list the *actions* derived from the original tasks propose by HOI4D, as well as the corresponding instance counts per action and object category in the training and test set of our HOI4D instance set, in Table 7. For H2O, we follow the official action split provided by the authors.

Table 7: Actions for the HOI4D instance split. Our proposed action-centric *instance split* for the HOI4D dataset, assigning each instance of a class to either the train or the test set for each object and action. In this work, we train and evaluate on the objects *bottle, bucket, kettle, knife, laptop, mug, pliers, scissors, stapler* from the HOI4D dataset.

| Action | Original Task(s) | Object Categories | Instances in Training Set | Instances in Test Set |
|---|---|---|---|---|
| bind | bind the paper | stapler | 38 | 11 |
| clamp | clamp something | pliers | 27 | 9 |
| cut | cut something | scissors | 40 | 12 |
| fill | fill with water by a kettle | mug | 24 | 7 |
| open/close | open and close the display | laptop | 45 | 13 |
| pour | pour all the water into a mug | bottle | 30 | 8 |
| | pour water into a mug | kettle | 40 | 12 |
| | pour water into another mug | mug | 24 | 7 |
| | pour water into another bucket | bucket | 38 | 11 |
| slice | cut apple | knife | 34 | 9 |

## A.4 HAND DETECTION

Detecting the human hand in the input image, as we wish to do so as to crop to the vicinity of the manipulated object, is prone to various types of errors, including failure to detect the hand (false negative) as well as incorrectly detecting the hand (false positive; e.g. detecting a hand of the wrong side).

We report the *detection rates*, i.e. the percentage of frames in which a hand is found, of two different approaches to detecting the hand in Table 8. Specifically, we consider the idea of using the hand pose reconstruction method HaMeR (Pavlakos et al., 2024) to detect the wrist keypoint of the hand in the image. Note that this method provides an estimated hand side along with the hand keypoints. We further investigate the use of obtaining a hand bounding box by grounding with Florence-2 (Xiao et al., 2024), conditioned with the prompts "human hand" (HOI4D) and "right hand" resp. "left hand" (H2O). We further investigate how well each approach distinguishes hand sides in Table 9.

Table 8: Detection rates of different approaches to locating the human hand in the input image. The Florence-2 (Xiao et al., 2024) + SAM-2 (Ravi et al., 2024) based approach detects hands in all images, yet struggles with correctly inferring hand sides.

| Approach | HOI4D | | H2O | | | |
|---|---|---|---|---|---|---|
| | Training | Validation | Training | | Validation | |
| | Det. Rate R | Det. Rate R | Det. Rate L | Det. Rate R | Det. Rate L | Det. Rate R |
| HaMeR (Pavlakos et al., 2024) | 95.91% | 92.17% | 93.31% | 94.37% | 98.36% | 100.00% |
| Florence-2 (Xiao et al., 2024) | 100.00% | 100.00% | 100.00% | 100.00% | 100.00% | 100.00% |

Table 9: Statistics of true positive (TP), false positive (FP) and false negative (FN) right hand detections on the H2O validation set (122 samples) for different approaches. The HaMeR (Pavlakos et al., 2024)-based method is reliable in its prediction of the hand side, while the Florence-2(Xiao et al., 2024)–based approach cannot distinguish left from right hands, leading to a high number of false positives and false negatives.

| Approach | TP | FP | FN |
|---|---|---|---|
| HaMeR (Pavlakos et al., 2024) | 121 | 1 | 1 |
| Florence-2 (Xiao et al., 2024) | 77 | 88 | 45 |

As evident from the results, Florence-2 achieves full recall on both training and validation splits of both datasets, while HaMeR consistently detects hands in about 90%-100% of all images. On HOI4D, where we consider interactions involving only the right hand, we can achieve a perfect detection rate by using Florence-2, or alternatively using HaMeR and then Florence-2 on images where HaMeR did not give hand detections. On the bimanual H2O, using Florence-2 will lead to more uncontrolled results, as the model frequently confuses left and right hands. HaMeR is able to detect all right hands, and almost all left hands on H2O's validation set. To always know to which region to crop, we hence opt to use the region surrounding the right hand based on HaMeR's wrist position prediction when performing inference on H2O.

## A.5 HYPERPARAMETERS

We use $T = 1000$ denoising steps for all diffusion-based models. For our guidance term, we experimented with different schedules and magnitudes of the guidance scale and empirically found a constant scale of 7, applied after the initial 100 denoising steps, to lead to the best results.

All models are trained for 250 epochs. Consistent with the original implementations, ReMoDiffuse (Zhang et al., 2023b) and MotionDiffuse (Zhang et al., 2022b)-based models are trained using the Adam (Kingma, 2014) optimizer with a learning rate of $2 \cdot 10^{-4}$, as well as a batch size of 128. MDM (Tevet et al., 2022b)-based models use a learning rate of $1 \cdot 10^{-4}$ and a batch size of 64.

## A.6 LLM USAGE

Large language models (LLMs) were employed as auxiliary tools to help refine text, polish wording, correct minor typos and improve clarity of the paper.

864
865
866
867
868
869
870
871
872
873
874
875
876
877
878
879
880
881
882
883
884
885
886
887
888
889
890
891
892
893
894
895
896
897
898
899
900
901
902
903
904
905
906
907
908
909
910
911
912
913
914
915
916
917

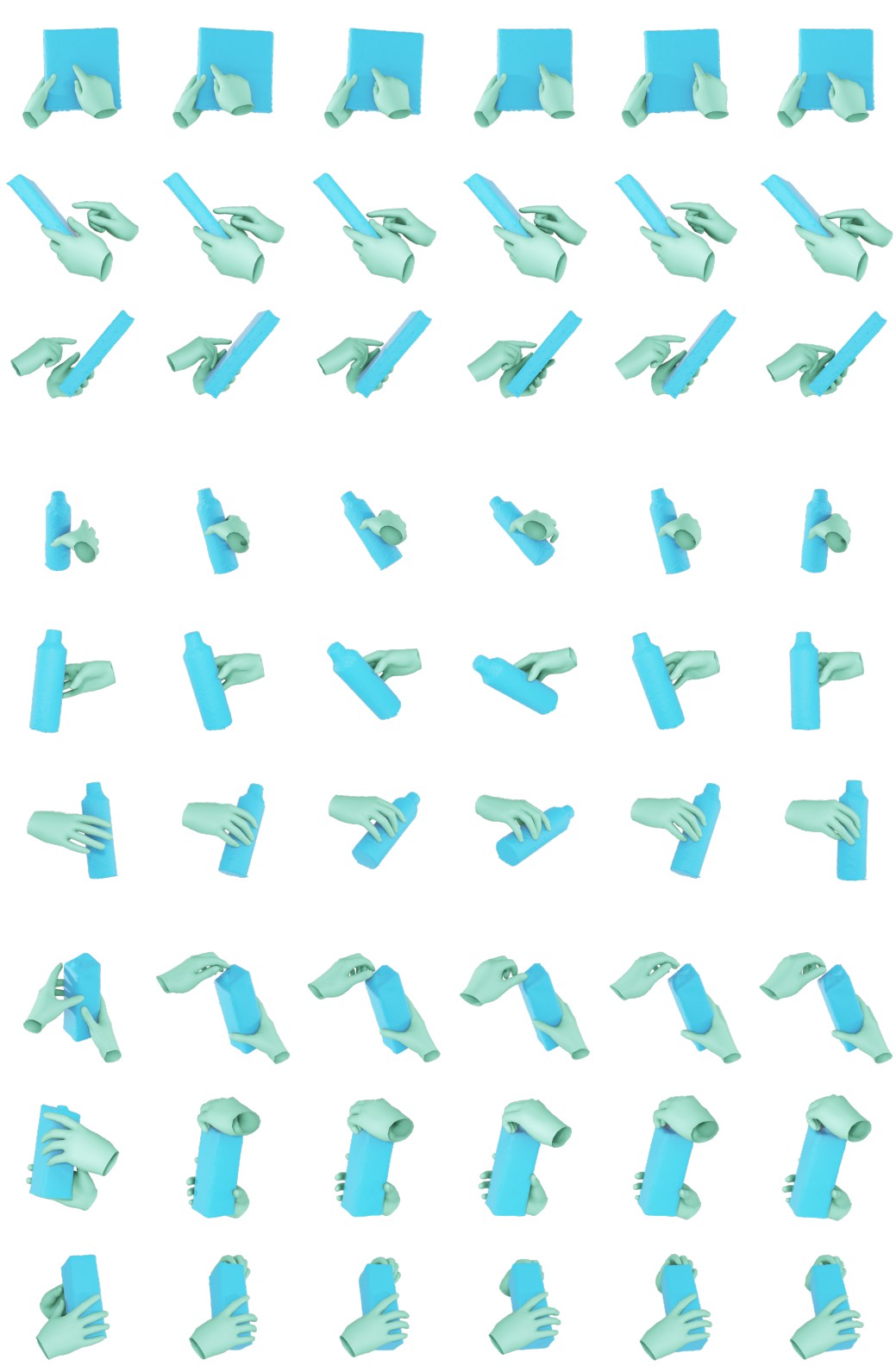

Figure 4: Further examples of hand-object interaction trajectories generated by our method, as viewed from different perspectives. Video files with further visualizations are also available as part of the Supp. Mat.

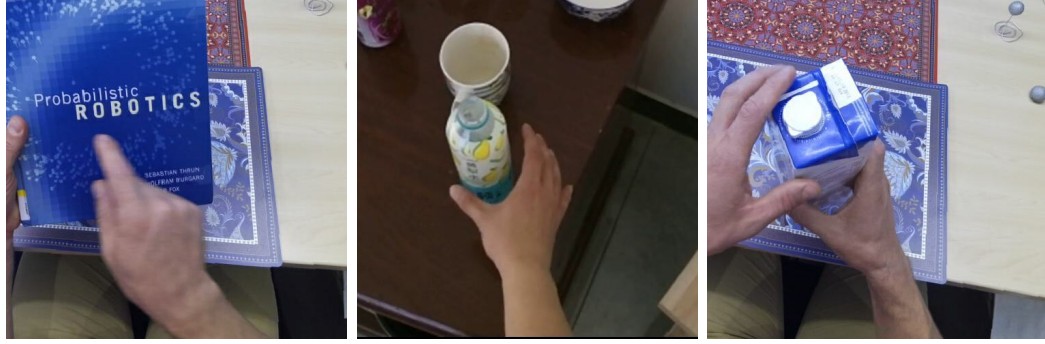

Figure 5: Conditioning images used to generate the trajectories in Figure 4 (top to bottom). The associated action labels are *read book*, *pour bottle*, *open milk*.

