# OpenReview forum: "SIGHT: Synthesizing Image-Text Conditioned and Geometry-Guided 3D Hand-Object Trajectories"
_ICLR.cc/2026/Conference — Submitted to ICLR 2026_

### Official Review · Reviewer_akaj · 2025-10-31

**Soundness:** 1
**Presentation:** 2
**Contribution:** 1
**Rating:** 2
**Confidence:** 4

**Summary:**

Authors proposed to generate future 3D hand-object interaction trajectories given a single RGB image and text prompt. Authors proposed a diffusion-based generative pipeline for solving the problem. The pipeline retrieves a 3D object from a database which is similar to the object contained in the input image. Then, the diffusion process is applied to synthesize future trajectories. During the process, interpenetration and contact consistency is further considered. Experiments on HOI4D and H2O demonstrate good performance compared to alternative approaches.

**Strengths:**

The paper is well structured and easy to understand what authors have done.

The problem tackled i.e. hand-object interaction motion generation in general seems like an important problem.

**Weaknesses:**

Less clear motivation: Authors proposed the SIGHT which generates trajectories of hands and objects given images and text prompts. However, this could be easily achieved by combining pre-existing text-to-hand motion generation (such as Text2HOI and DiffH2O) and 3D object reconstruction pipelines. I think authors need to empirically compare their method to these simple baselines. More importantly, besides the difference in the scenario, there is not much enough contributions in the technical side.

Metric seems not sufficient: Future trajectories could not be properly measured by the reported measures such as FID. diversity, etc. I think there should be more semantic measures involved to properly evaluate whether the generated motion achieves the intended prompt. Or, I think user studies need to be involved to properly evaluate the performance. In this form, it might be hard to judge the effectiveness of the methods for the intended task.

Lack of in-the-wild experiment: Since the scenario is challenging, it is required to check whether the method will generalize to in-the-wild scenes with clutter, unseen object geometries, or non-rigid objects. The current datasets (HOI4D, H2O) are controlled benchmarks with known household-like manipulation actions. I think authors may have to include more in-the-wild samples.

**Questions:**

The mesh retrieval mechanism is not well explained. How large and how diverse is the mesh database?

---

### Official Review · Reviewer_bFJx · 2025-11-01

**Soundness:** 3
**Presentation:** 3
**Contribution:** 2
**Rating:** 4
**Confidence:** 5

**Summary:**

1. This paper introduces a 3D hand–object motion generation framework that leverages an image modality along with a brief textual description.

2. A diffusion guidance method is proposed at inference time to enforce physically plausible motions.

3. An image-based retrieval approach is proposed to retrieve object meshes during inference.

**Strengths:**

1. The paper introduces a new task of generating 3D hand–object interaction trajectories from a single image and a text prompt.

2. The authors present various experiments and impressive video visualizations.

3. The proposed method demonstrates superior performance compared to other baselines.

4. The authors plan to publicly release the code and model parameters.

**Weaknesses:**

1. The generated motions do not look realistic in the video results.
For example, the wrist and object rotations are not aligned. In the “pour bottle” case from the baseline comparison in the supplementary video, after pouring and setting the bottle upright, the bottle and wrist appear to rotate independently. This behavior looks similar to the MotionDiffusion method as well. The “pour bottle” case in the generated trajectories shows a similar issue.

2. The objects appear to move on their own.
For instance, in the “grasp chips” case from the baseline comparison in the supplementary video, before the hand grasps the chips, the object moves as if magnetic forces are applied.

3. The image information does not seem to provide clear benefits.
In practice, utilizing it effectively is difficult. However, it could serve as a strong prior for initializing the hand and object positions, or for reconstructing the 3D object mesh using HOLD [1]. I believe humans would prefer text-only methods; therefore, such models should demonstrate better performance and controllability (similar to W4).

4. The initial hand and object poses are not aligned with the input image.
The model does not seem to properly interpret the input image information. For example, in the supplementary videos (open lotion, grab lotion), the initial poses of the hand and object do not match the input image, indicating poor alignment at the start of the motion.
In the “pour milk” case, the grasp point on the object differs from the image. In the image, the hand grasps the front and back of the milk carton, whereas the generated motion touches its sides.

5. In addition to W4, a metric measuring the alignment between the initial generated poses and the input image is needed.
Since ground-truth motion data is available, you could measure the Chamfer Distance for the object, MPJPE for the hand, and relative distance and orientation errors for both the hand and the object.

6. Why didn’t you compare with Text2HOI [2]?
I believe this method was trained on the H2O dataset, and its results look realistic.


[1] Fan, Zicong, et al. "Hold: Category-agnostic 3d reconstruction of interacting hands and objects from video." Proceedings of the IEEE/CVF Conference on Computer Vision and Pattern Recognition. 2024.

[2] Cha, Junuk, et al. "Text2hoi: Text-guided 3d motion generation for hand-object interaction." Proceedings of the IEEE/CVF Conference on Computer Vision and Pattern Recognition. 2024.

**Questions:**

1. Clarification on L_pen:
It is unclear what parameters L_pen optimizes — does it optimize the noise, or the pose and translation parameters?

2. Choice of CLIP and SigLip:
Why is CLIP used for text and SigLip for image features? Since both are variants of CLIP, the motivation for using different encoders should be clarified.

3. Missing citation for diffusion models (L220):
There is no citation provided for state-of-the-art diffusion models mentioned around line 220. Please include relevant references.

4. Notation inconsistency (L160, L273, L330):
The symbol M is used both for the hand–object interaction generator (L160) and for the 3D object (L273), which may cause confusion. Additionally, the motion generation method symbol M in L160 and L330 is not consistently aligned.

---

### Official Review · Reviewer_N5zj · 2025-11-01

**Soundness:** 3
**Presentation:** 3
**Contribution:** 1
**Rating:** 4
**Confidence:** 2

**Summary:**

This paper proposes a novel task, SIGHT, which aims to generate realistic hand–object motion conditioned on a single image depicting the starting frame of the interaction and a text-based description of the future action. This contrasts with prior works that considered only text conditions or assumed that the 3D object geometry was already given. To tackle this task, the paper introduces SIGHT-Fusion, a diffusion-based image–text conditioned motion generation model with inference-time guidance that enforces physically plausible interactions by penalizing shape penetrations. In experiments, it outperforms the designed baselines in both the diversity and quality of the generated motions.

**Strengths:**

**(1) Novelty of the proposed task**

This paper proposes a novel task, SIGHT, which focuses on hand–object interaction motion generation grounded on a single image depicting the starting frame of the interaction and a text-based description of the future action. I believe the proposed task is novel and potentially important for the robotics field.

**(2) Presentation quality**

The paper is well written and easy to understand. The overall presentation quality is good.

**Weaknesses:**

**(1) Missing relevant baselines in the comparisons**

In the comparisons, the proposed method is only evaluated against an existing human body motion generation model, whereas there are many hand–object interaction generation models (e.g., those discussed in Sec. 2) that are much more relevant in terms of both task and technical similarities. The proposed techniques (e.g., joint hand and object trajectory denoising or test-time guidance to avoid interpenetration) are commonly used in several existing diffusion-based models mentioned in the "Hand–object interaction synthesis" paragraph in Sec. 2, so comparisons against these would be most important. Since the proposed framework obtains explicit 3D object geometry as an intermediate representation, e.g., comparing against a variant of the Text-to-HOI baselines with the given 3D object geometry seems straightforward.

**(2) Misleading explanation of prior works**

In L63–66, it is stated that existing methods “do not generate trajectories” of hand–object interactions; however, prior works (as later mentioned in L139–141), including those based on reinforcement learning, do address trajectory generation.

**(3) Physically implausible generation results in Fig. 1**

The top row of Fig. 1 shows hand–object interaction motions that are physically implausible (e.g., the laptop is articulated by a non-contacting hand). This raises further questions about the superiority of the proposed method compared to reinforcement learning–based methods, which can already incorporate action conditions while generating physically plausible motions.

**Questions:**

(1) Does the framework use separate models for single-hand vs. two-hand interactions, and for rigid vs. articulated objects?

(2) Recent grounded generation methods typically evaluate alignment with the grounding information (i.e., the text description and the first-frame image in this work). I am wondering if it would be feasible to measure the alignment between the first-frame image and the generated motions as well.

---

### Official Review · Reviewer_HFMJ · 2025-11-01

**Soundness:** 2
**Presentation:** 2
**Contribution:** 1
**Rating:** 2
**Confidence:** 4

**Summary:**

The paper proposes SIGHT, a diffusion-based framework for generating 3D hand–object interaction trajectories from a single RGB image and text description. The method integrates visual–text conditioning, retrieval-based object meshes, and geometry-guided diffusion to ensure physically plausible and semantically consistent motions. Experiments on HOI4D and H2O show that SIGHT outperforms existing motion diffusion baselines in accuracy, FID, and contact quality, demonstrating improved realism and reduced interpenetration.

**Strengths:**

- The paper is well-organized

**Weaknesses:**

- I find the task setting somewhat unclear. Without the explicit use of real object meshes during generation, it is difficult to understand how the model ensures spatially accurate contact regions. To be honest, I am not convinced by the retrieval module — it seems to introduce additional uncertainty rather than guaranteeing accurate geometry, especially for unseen object. Moreover, it is unclear how the retrieved mesh is matched to the real object in terms of scale and alignment in 3D space, and how it is correctly integrated into the generated motion trajectory. This part of the framework remains confusing.
- Why are baselines such as MDM, MotionDiffuse, and ReMoDiffuse—which are primarily designed for full-body human motion generation—used for comparison in this task, rather than diffusion-based models specifically tailored for hand–object interaction generation? A justification for this baseline choice would help clarify the experimental setup and strengthen the fairness of the comparison.
- Most hand–object interaction generation methods, whether conditioned on object geometry or with additional textual descriptions, need to explicitly or implicitly learn true affordance or contact relationships to produce physically reasonable interactions. In contrast, the experimental setting in this paper seems somewhat biased — since the input image (the hand and object are almost in contact) inherently leaks contact and affordance cues, the task becomes substantially easier. As a result, the technical novelty appears limited, and the method does not meaningfully address more challenging or general human–object interaction scenarios.

**Questions:**

Please refer to weakness.

---

### Meta-Review · Area_Chair_qKLV · 2026-01-06

**Summary:**

This paper presents SIGHT, a diffusion-based framework for generating future 3D hand–object interaction trajectories from a single RGB image and a text prompt, targeting a setting where the object geometry is not given. At inference time, it retrieves a plausible 3D object mesh from a database based on the input image, then synthesizes hand–object motion with image–text conditioning and geometry-guided diffusion, including guidance terms that promote physically plausible interactions by discouraging interpenetration and improving contact consistency. Experiments on HOI4D and H2O show that the method outperforms motion-diffusion baselines in motion quality/diversity and contact-related metrics, producing more realistic interactions with reduced penetration.

**Reviewer Concerns:**

Reviewers raised the following main concerns:
1. Unclear task framing and novelty: Reviewers questioned whether the setting is well-defined and sufficiently novel, noting it may be achievable by combining existing text-to-HOI generation with object reconstruction.
2. Retrieval + geometry integration is unclear: The retrieval module may add uncertainty, and the paper does not clearly explain how the retrieved mesh is scaled/aligned to the real object and integrated to ensure accurate contact, especially for unseen objects.
3. Baseline choice and fairness: Comparisons rely heavily on full-body motion diffusion baselines, while more relevant hand–object interaction methods (e.g., Text2HOI/DiffH2O and other HOI diffusion/RL approaches) are missing or insufficiently justified.
4. Evaluation is limited: Current metrics are not sufficient to assess physical plausibility and semantic correctness; reviewers asked for stronger quantitative measures (including image-to-initial-pose alignment), more thorough comparisons, and/or user studies.
5. Qualitative realism and generalization concerns: Videos show physically implausible behaviors and weak use of image conditioning, and there is limited evidence of robustness/generalization beyond controlled datasets (with potential bias from near-contact input images).

**Reviewer Scores:**

This submission received all negative scores (2, 4, 4, 2). The authors did not submit a rebuttal. Most reviewers raised concerns about the evaluation setup and the fairness of the comparisons.

---

### Decision · Program_Chairs · 2026-01-26

Reject